# Super-Resolution integrated Semantic Segmentation Method for the Corner Position of Catenary Bolt

1st Yilin Chen
School of Electrical Engineering
Southwest Jiaotong University
Chengdu, China
chenyilin@my.swjtu.edu.cn

2nd Minyang Wei
School of Electrical Engineering
Southwest Jiaotong University
Chengdu, China
weimy@my.swjtu.edu.cn

3rd Junjie Ma
School of Electrical Engineering
Southwest Jiaotong University
Chengdu, China
772031448@qq.com

4th Na Qin
School of Electrical Engineering
Southwest Jiaotong University
Chengdu, China
qinna@swjtu.edu.cn

5th Deqing Huang
School of Electrical Engineering
Southwest Jiaotong University
Chengdu, China
elehd@home.swjtu.edu.cn

*Abstract*—Locating the corner position of a hexagon bolt with high precision is crucial, especially when dealing with low-resolution images during the tightening of wrist bolts in overhead contact systems. We present a novel framework for an SR-integrated semantic segmentation method. In this paper, we select the high-performance SRGAN model for low-resolution image reconstruction; then improve the Deeplabv3+ segmentation model to achieve efficient segmentation of hexagonal bolts; and finally determines the bolt corner positions based on Line Segment Detector. The novelty of this method lies in integrating image super-resolution into the semantic segmentation model, followed by the improvement of the segmentation network using the lightweight MobileNetv2 backbone, ultimately achieving precise bolt corner location. Experiments demonstrate that the proposed method improves corner detection accuracy by 38.93% compared to the original low-resolution method across different scenarios, proving its practical engineering significance.

*Index Terms*—Corner location of bolt, Semantic segmentation, Super-resolution, Catenary arm

## I. Introduction

The catenary system is a key component of modern rail transportation, providing continuous electric energy for high-speed trains and promoting the rapid development of railway systems. However, the bolts on the catenary arm frequently loosen due to adverse weather and prolonged pantograph-catenary contact, affecting train stability and safety. Currently, bolt tightening primarily relies on manual operations, requiring maintenance personnel to manually tighten bolts in outdoor environments. This process is not only inefficient but also poses high safety risks and costs. The Fig.1 shows the catenary arm and the bolts compoment. Therefore, it is crucial to use visual sensors and robotic arm automation technology to achieve fast and accurate bolt tightening[1-2].

Image-based visual servoing (IBVS) technology achieves precise positioning of the robot end-effector by analyzing the image features of the target object, such as feature points, lines, and planes[3-4]. In this study, when the robotic arm moves near the parts to be tightened on the catenary arm, the camera captures the hexagonal feature of the bolt—the six corner positions—and matches it with the standard template to control the robotic arm to accurately fit the socket onto the bolt and complete the tightening. However, in actual maintenance, the resolution of bolt images is often low, leading to inaccurate corner position detection, which affects the positioning and operation of the robotic arm. Therefore, obtaining high-precision corner position information from Low-Resolution(LR) images is the core issue of this study.

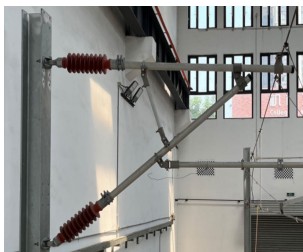 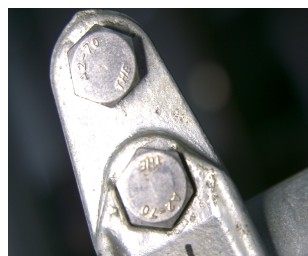

(a)The catenary arm.    (b)Bolt compoment.

Fig. 1: The picture of the catenary arm and the bolts compoment.

Currently, bolt research mainly focuses on detecting bolt loosening, with limited research on accurately extracting the six corner positions of the bolt. Nonetheless, several studies have attempted to extract bolt contour information using traditional image processing techniques, such as edge detection, thresholding segmentation, and shape processing. For example, Wang et al.[5] used an object detection algorithm to locate remote bolt areas, followed by adaptive thresholding segmentation and Hough transform technology[6] to obtain the contour and

position information of hexagonal nuts. Luo et al. [7] used thresholding segmentation to extract the hexagonal nut subgraph from the bolt connection graph and identified each nut's edge line using a Canny edge detector. However, these traditional detection techniques are mainly suitable for situations with significant color contrast between the hexagonal bolt and the background. They are less robust and susceptible to external light, angle, and noise. Additionally, these methods are affected by subjective parameter settings, limiting their reliability and stability in practical applications. Therefore, developing more accurate and robust bolt detection techniques to adapt to complex environments and improve automated maintenance efficiency is particularly important.

With technological advancements, the field of computer vision and deep learning has witnessed significant developments. With its excellent feature extraction capabilities, Convolutional Neural Networks (CNN) have achieved breakthrough results in key areas such as Semantic Segmentation(SS) and Super-Resolution(SR). The advent of Fully Convolutional Networks (FCN) [8] revolutionized the traditional segmentation approach. By removing the fully connected layer in traditional CNNs and fully utilizing the convolutional layer, FCNs achieved leading SS results at the pixel level. This innovation inspired a series of high-performance segmentation network models, including SegNet, DeconvNet, the DeepLab family, U-Net, and PP-Liteseg based on the PaddlePaddle framework [9-13]. These advanced algorithms can effectively and clearly segment hexagonal bolts from catenary arm components.

Field-collected images often suffer from insufficient resolution, impacting the accuracy of manual labeling and model train for predictions. To tackle this, SR is essential, as it enhances image clarity, allowing convolutional neural networks (CNNs) to extract detailed features and learn the mapping from LR to high-resolution (HR) images, thus aligning training and testing datasets more closely.Dong et al. pioneered the SRCNN algorithm [14], which quickly reconstructs high-quality HR images with a three-layer convolutional model. They further refined this with the FCN-based FSRCNN [15], offering a more efficient and accurate reconstruction method. Despite these improvements, there's still room for refining detail and texture processing.In recent years, Generative Adversarial Networks (GANs) have become prominent in SR tasks, with their discriminators adept at learning complex loss functions. Ledig et al. [16] introduced SRGAN, the first GAN application to single image SR, which effectively restores realistic textures, vital for detecting and maintaining minor damages on hexagonal bolts on catenary arms over time

To summarize, we present our novel framework for an SR integrated bolt SS method. During maintenance, we overcame the resolution limitation by generating corresponding HR images through the SRGAN model in the face of LR images. On this basis, we combined

the improved Deeplabv3+ model to complete the SS of hexagon bolts and accurately position the corner points of the contact line arm bolts. Compared with manual maintenance and traditional vision methods, the proposed method has the following significant advantages:

Firstly, this method improves bolt extraction effectiveness and robustness through the pre-processing of SRGAN image reconstruction based on data enhancement technology. It also effectively addresses common fuzzy and LR image problems during overhead catenary arm maintenance. Images with a single data source and low quality often face robustness and accuracy issues in manually labeled datasets in diverse environments. Therefore, improving the overall quality of images lays a solid foundation for the subsequent SS of hexagonal bolts.

Secondly, in selecting the SS model, this paper integrates the lightweight MobileNetv2 backbone into the Deeplabv3+ model to enhance the SS effect. Compared with traditional models, the improved Deeplabv3+ model reaches new heights in segmentation accuracy and overall performance. This combination optimizes the model's computational efficiency and significantly improves segmentation accuracy.

Furthermore, boundary detection of the separated hexagon bolt contour is performed to determine the order of adjacent boundaries, obtaining the final position of the hexagon bolt intersection. The high-precision Line Segment Detection (LSD) [17] is used to detect the edge segment of the mask image, eliminating image edge unsmoothness errors. By expanding the dataset of HR images of different parts of the contact line arm bolt collected in the laboratory, experimental results show that the proposed method achieves a high level of experimental effect, with a detection accuracy increase of 38.96%. This result highlights the critical role of SR in improving the performance of LR bolt image segmentation, benefiting the maintenance of the catenary arm by mechanical arms.

The structure of this paper is arranged as follows: Section I introduces the background of overhauling the contact net wrist arm, the visual servoing technology of the robotic arm based on image visual servoing, and the challenges it faces. Section II provides a detailed overview of the design of the bolt corner detection model integrating SR into SS. Section III describes and analyzes the preparation of experimental data, process, and results. Finally, Section IV concludes the paper.

## II. Design of Detection Model

### A. Super-Resolution Image Reconstruction based on SRGAN Model with Image Data Augmentation

When examining the performance of SS models, it is essential to address the significance of datasets preparation, which underpins the model's learning and generalization capabilities. A high-quality datasets should be based on diverse shooting conditions and accurate manual annotation. However, in the practical task of

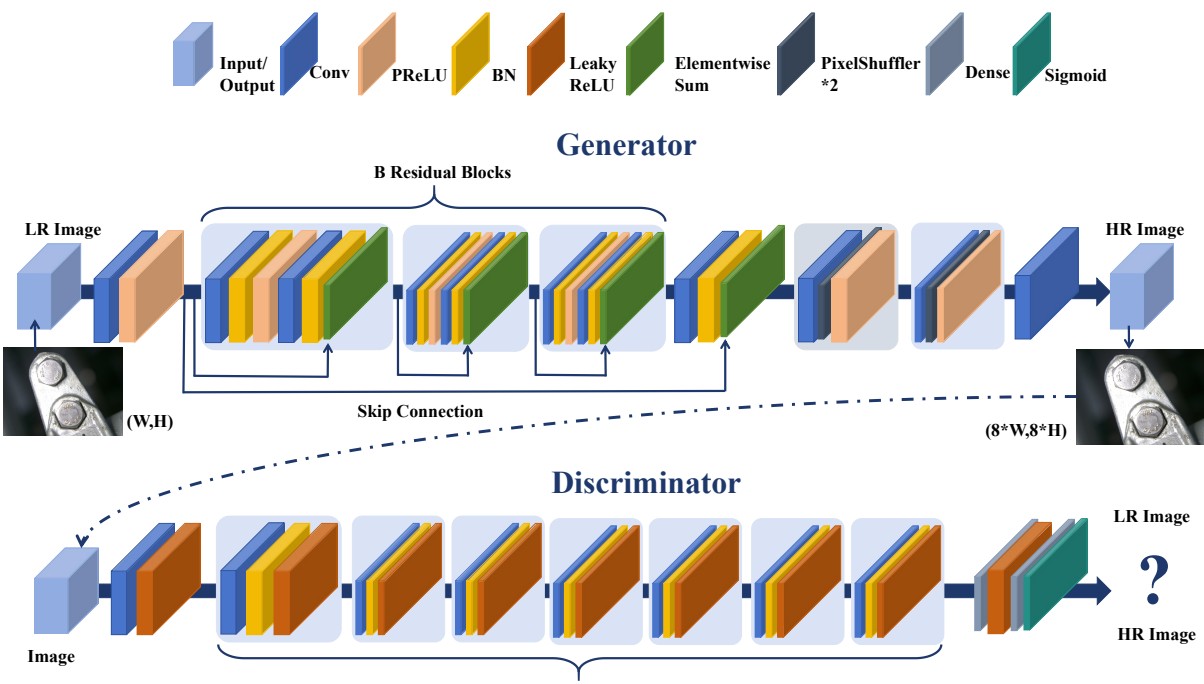

Fig. 2: The structure of SRGAN network.

tightening bolts on the catenary arm, LR images are often encountered. This not only increases the data's uniformity but also complicates manual annotation, thereby affecting the model's training efficacy and generalization ability. To address these issues, integrating image SR into image data enhancement is particularly crucial.

Image data augmentation technology plays a pivotal role in the realms of machine learning and deep learning. This technique involves applying a series of carefully designed transformations to the original image, including but not limited to rotation, translation, scaling, flipping, and noise addition. These transformations create new images with subtle differences while retaining the original features. These newly generated images are used as additional training samples along with the original dataset. Data augmentation not only expands the training dataset's breadth and enhances the model's learning potential but also simulates various outdoor environments such as sand, rain, night, and bright light. By introducing diverse image transformations, the model is exposed to richer data forms during training, resulting in stronger adaptability to diverse environmental data during actual catenary arm maintenance. This effectively reduces the risk of model overfitting and significantly improves the model's generalization and robustness.

Image SR further strengthens the model's data foundation and enhances the accuracy of data labeling by improving the details of LR images. The SRGAN network, as a representative of image SR , adeptly combines deep learning and adversarial training. The Generator aims to produce SR images indistinguishable from real HR images, while the Discriminator continually optimizes its discrimination ability. Throughout this process, the Generator gradually improves, ultimately generating high-quality images that enhance the reality and detail of the image, effectively addressing the smoothing problem inherent in traditional algorithms. The structure of the network is depicted in the Fig.2.

1) Generator

The Generator consists of multiple consistent residual blocks, each containing two convolutional layers. The 3×3 convolution kernel and 64 feature maps effectively capture global and local image features. The introduction of batch normalization and the ReLU activation function further enhance the model's learning ability and generalization. The sub-pixel convolutional layer at the network's end achieves precise image magnification through channel expansion, providing clearer image results and showcasing deep learning's potential and excellent performance in SR reconstruction.

2) Discriminator

The Discriminator's architecture employs the Leaky ReLU activation function and avoids Max pooling to preserve image details. Inspired by the VGG network, the following steps are enhanced: composing multiple convolutional layers, increasing the number of filter kernels, emphasizing detailed local feature inspection, and improving computational efficiency through stride convolution. Finally, the feature map is input into the dense layer, and the final classification probability is output by the

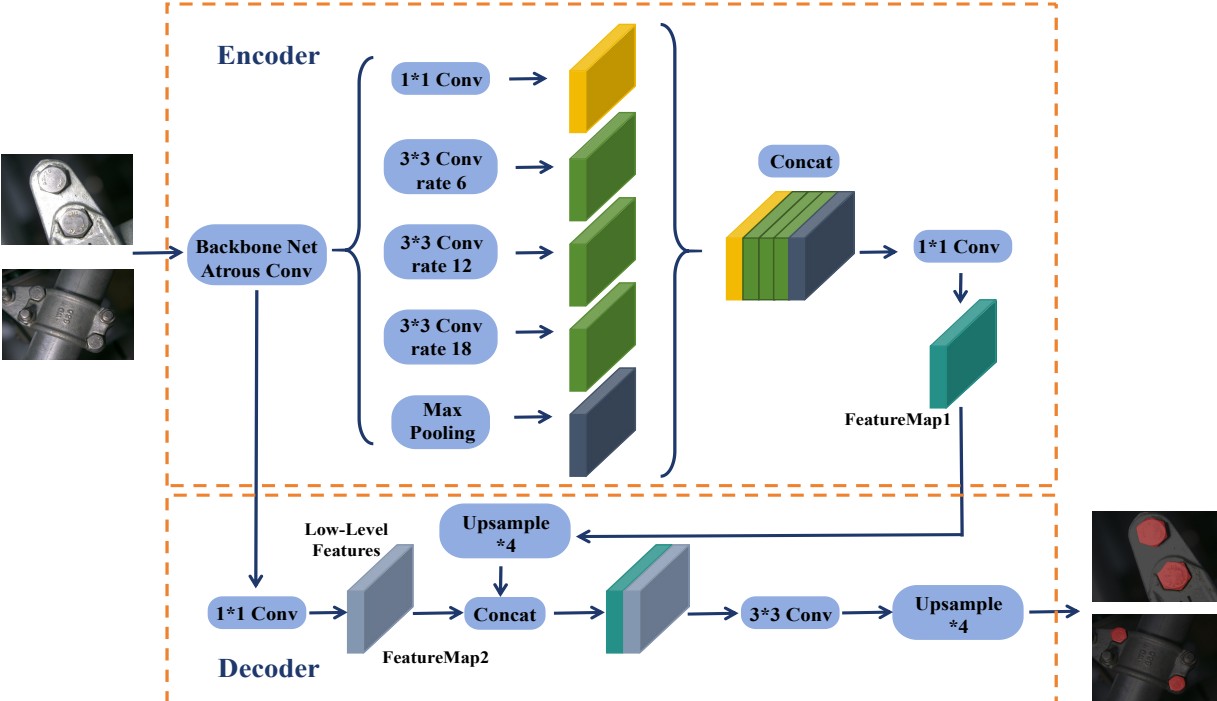

Fig. 3: The structure of Deeplabv3+-Mobilenetv2 network.

sigmoid function, accurately distinguishing between real and fake images, thus providing a solid foundation for SR technology.

3) Network usage

Thanks to the collaborative mechanism of the SRGAN's Generator and discriminator, only a set of real HR images need to be provided initially. The network generates corresponding LR images through its down-sampling process and learns the complex mapping relationship between HR and LR images. In practical applications for catenary arm maintenance, only the generator module of SRGAN and its trained weights are deployed. This not only simplifies the actual network model's scale to meet lightweight standards but also rapidly recovers HR images with rich details by processing LR images captured by the manipulator's front camera. This provides a solid foundation for the subsequent contour segmentation task of hexagonal bolts.

B. Improved Deeplabv3+ Semantic Segmentation Model

1) DeepLabv3+ Network Concept and Framework

The DeepLab family of advanced deep learning image SS models effectively combines low-level visual features and high-level semantic information to achieve more accurate image segmentation. As an optimized version of the series, DeepLabv3+ introduces an Encoder-Decoder structure to perfectly fuse multi-scale information. The Encoder component manages feature extraction at different resolutions to balance accuracy and time consumption. The network framework schematic Fig.3 demonstrates this innovative design.

2) Backbone Network Innovation

With the continual advancement of CNN technology, Depthwise Separable Convolution(DSC) has become a typical lightweight network structure due to its low parameter count and operational cost. The backbone network, responsible for extracting image features, directly affects model performance and computational complexity. Although the original DeepLabv3+ uses Xception as the backbone, which offers optimization, it still has high computational complexity and parameter count. To meet

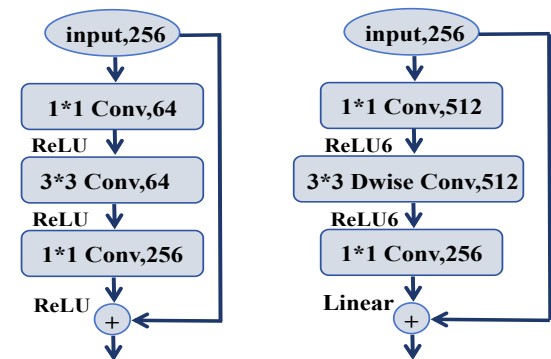

Fig. 4: The structure of inverted residual module.

the requirements of bolt corner location in catenary arm maintenance, we innovatively use MobileNetV2 as the backbone network for fast and efficient feature extraction. MobileNetV2's inverted residual module initially expands the number of channels through 1×1 pointwise convolution, then performs DSC, and finally reduces the dimension to minimize feature information loss. This structure, which was introduced early in the network, captures the richness of features that are critical to the

SS task, and its structure is shown in Fig 4.

Comprehensive evaluations in the Table I show that using MobileNetV2 as the backbone network offers significant computational complexity advantages without compromising segmentation performance. This supports deploying efficient SS models in computing resource-constrained environments.

TABLE I: Model Parameters and Performance Metrics

| Model | Parameters$(M)$[a] | MACs$(HM)$[b] | Size$(MB)$ |
|-------|------------------|-------------|----------|
| MobileNetV2 | 3.4 | 300 | 15 |
| Xception | 22.9 | 11700 | 70-80 |

[a] Number of parameters in the backbone network.
[b] Number of parameters in the backbone network.

### 3) Encoder Core Innovation

In the bolt images captured during maintenance, the target bolt occupies most of the area, requiring a large receptive field from the model. The core of DeepLabv3+ introduces dilated convolutions, allowing each convolutional layer to output a broad range of pixel information without losing detail. Dilated convolutions extract effective features across pixel levels. DeepLabv3+ uses multiple parallel Atrous convolutions (ASPP) with different rates to extract features at varying resolutions and encode rich semantic information.

### 4) Loss Function Definition

The SS task involves classifying each pixel of the input image. Network training progresses by comparing predicted results with class labels in the actual segmentation map. Traditional DeepLabv3+ uses a composite loss function consisting of Cross-Entropy loss and Auxiliary loss. The Auxiliary loss is calculated on multi-layer feature maps to improve performance at different resolutions and enhance small object detection. For this task, the classification category simplifies to bolt and background, with the bolt being a large target object. Therefore, the composite loss function replaces the auxiliary loss with the Dice loss for training.

- Cross-Entropy loss

The cross entropy loss measures the difference between the predicted probability distribution of the model and the actual label probability distribution, and is commonly used in multi-class classification. It is defined by the following formula:

$$l_{C-E} = -\sum_{i=1}^{N} y_i \log(p_i) \tag{1}$$

In the fomular: $N$ is the number of categories, $P = \{p_1, p_2, \cdots, p_N\}$ is the probability that a pixel belongs to the Cth category, and $Y = \{y_1, y_2, \cdots, y_N\}$ is the value in the one-hot encoding vector.

- Dice loss

Dice loss is designed for binary or multi-class classification problems. It measures the similarity of samples based on the Dice coefficient to deal with class imbalance. It is calculated as follows:

$$l_{Dice} = 1 - Dice = 1 - \frac{2\,|X \cap Y|}{|X| + |Y|} \tag{2}$$

In the fomular: $X$ and $Y$ represent the pixel values of the predicted and true labels respectively, the $Dice$ coefficient measures the similarity between the two, and smaller is better as a loss function.

### C. Hexagon Bolt Corner Positioning Method

This study focuses on using semantic segmentation algorithms on HR reconstructed images to accurately extract hexagonal bolt contours. Based on advanced SS techniques, we generate segmentation mask maps of hexagonal bolts. However, segmentation mask edges are often not smooth enough, and traditional edge extraction methods, such as the Canny operator, struggle to obtain ideal flat contours. Thus, we introduce the Line Segment Detector(LSD). Leveraging LSD's efficient line segment detection performance, we perform a series of screening and positioning operations on detected line segment information to obtain the six boundary lines of the hexagonal bolt, ultimately locating the bolt's corners. The Fig. 5 illustrates the LSD detection results and final corner position determination results.Below are the LSD operation steps and line segment screening and positioning steps:

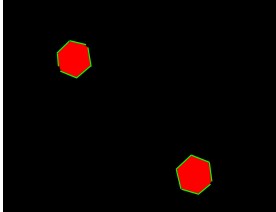 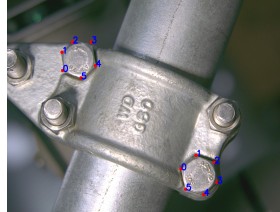

(a)LSD's handling of segmentation masks.  (b)The final result of corner position.

Fig. 5: The LSD detection results and final corner position determination results.

### 1) LSD Operation Steps:

Step 1: Image scaling pre-processing. Reduce the input image to 80% of its original size and apply Gaussian downsampling to minimize or eliminate jagged effects, preventing other interference noise.

$$G = \sqrt{\left(grad_x(x,y)^2 + grad_y(x,y)^2\right)} \tag{3}$$

$$\theta = \arctan\left[-\frac{grad_x(x,y)}{grad_y(x,y)}\right] \tag{4}$$

In the formula:$G$represents the gradient of the pixel at $(x,y)$ along the $x$ and $y$ axes; $\theta$ is the gradient angle at that point.

Step 2: Calculate and sort image gradient magnitude and row and column angles. A 22 template calculates gradient magnitude $G$ and row-column angles $\theta$. This small template speeds up calculation and ensures neighborhood direction distribution independence.

Step 3: Image gradient sorting and pixel filtering. Regions with sharp gradient magnitude changes are likely to contain strong edges. Sorting pixel gradient magnitudes enhances subsequent line segment calculation and detection accuracy. In LSD calculations, pixels with gradient amplitudes exceeding the threshold are filtered out.

Step 4: Region generation. Select unused pixels from the sorted pixel list as seed points. Search for unused pixels satisfying specified angle conditions in the 8-neighborhood of seed points to form the support line region.

2) Line Segment Screening and Positioning Steps

Step 1: Calculate and sort segment lengths. Use the Euclidean distance formula to calculate the lengths of all line segments in the mask image, then sort them. Identify the six longest line segments as the hexagonal bolt's boundaries.

$$Length = \sqrt{(x_1 - x_2)^2 + (y_1 - y_2)^2} \qquad (5)$$

Step 2: Sort endpoint coordinates. Since the obtained line segment information lacks order, further processing is required. Traverse endpoint coordinates using the Euclidean distance formula to find the closest endpoint pairs, determining the adjacent order of the six boundaries.

Step 3: Determine corner positions. Using the principle of the intersection of two straight lines, determine the six corner positions of the hexagonal bolt. Divide each corner point's coordinate values by 8 according to the reconstruction coefficient to obtain the hexagonal bolt corner positions in the resolution picture.

Using the above positioning method, we accurately identify the hexagonal bolt's boundary and corner positions, providing a solid foundation for subsequent image-based visual servoing of the manipulator. The figure below illustrates this flow, depicting the complete steps from image processing to corner location.

### III. Experiment

This section describes the implementation of the catenary arm hexagonal bolt corner position method, which incorporates SR into SS. The entire flowchart is depicted in Fig. 6. The process is divided into four parts: dataset preprocessing, image reconstruction using the SRGAN model, SS of hexagonal bolts based on an improved Deeplabv3+ model, and determination of the corner positions of the hexagonal bolts.

### A. Dataset Pre-processing

LR images with a resolution of 512×375 were collected during the overhead contact net overhaul. We used a manual zoom camera, Basler, to capture HR bolt images

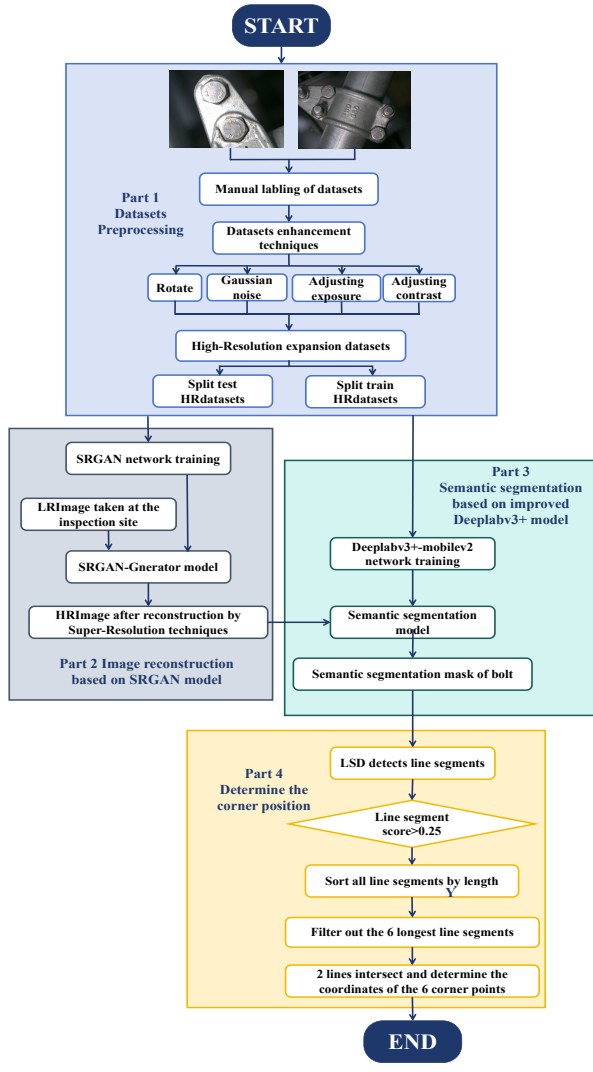

Fig. 6: The entire flowchart of the catenary arm hexagonal bolt corner position method.

with a resolution of 4096×3000 from the overhead contact net wrist arm model during the training data acquisition stage, after calculating a SR factor of 8. All images were downsampled (coefficient 8) to create LR image datasets with a resolution of 512×375. It is important to note that the LR datasets are used only as a control object for the subsequent SS network and not as an operational step of the method. Considering the actual outdoor working environment, which includes sand, rain, night conditions, and strong light irradiation, and given the limited number of captured data, data enhancement techniques such as rotation, horizontal cropping, Gaussian noise, and exposure adjustment were applied to both datasets. After these enhancements, two groups of 1400 HR and LR experimental datasets were obtained and randomly divided in an 8:2 ratio. The number of images of catenary arm bolts used for SRGAN and Deeplabv3+ network training is 1120, while 280 images were used for testing. Part of the catenary arm bolt image datasets

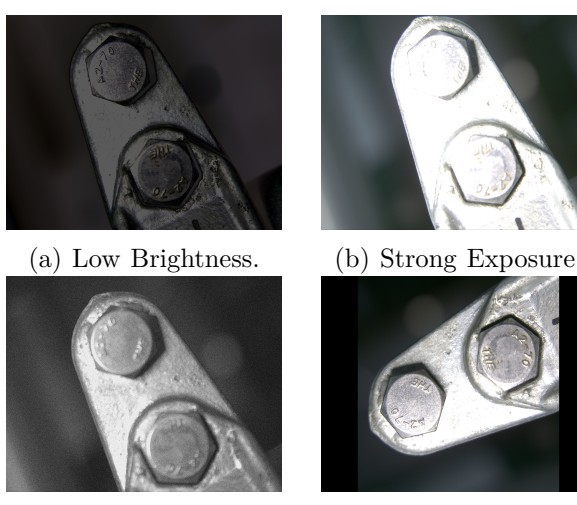

| (a) Low Brightness. | (b) Strong Exposure. |
|---|---|

| (c) Gaussian Noise. | (d) Rotate. |
|---|---|

Fig. 7: The display of enhanced datasets.

enhanced by data augmentation techniques is shown in Fig. 7.

### B. Image Reconstruction for SRGAN Model

1) Datasets Selection and Model Training: During the SRGAN model train phase, we selected HR datasets with a resolution of 4096×3000. The model can automatically downsample the input HR image by a factor of 8 through bicubic interpolation, simulate the characteristics of LR images, and finally reconstruct a HR image based on the actual LR image. Since the semantic segmentation network is the key model technology of the whole method, and image SR is a promotional step, we only train HR image reconstruction on the SRGAN model, which is configured with an Ubuntu 18.04 system and performed on a 2×22G RTX 2080Ti GPU device. The reconstruction factor is 8, with default epochs set to 200, Batch size set to 32, momentum set to 0.9, and an initial learning rate of 0.0001.

2) Evaluation Indicators:

- $PSNR$(Peak Signal-to-Noise Ratio)

Measures image quality by comparing the difference between the original and SR images through the Mean Squared Error (MSE) of each pixel value. It is calculated as follows:

$$PSNR = 10 \lg \left[ \frac{(MAX_I)^2}{MSE} \right] \tag{6}$$

In the formula: $MAX_I$ is the maximum possible pixel value of the image(255 in this paper); The expression for the $MSE$ is given as follows. A higher $PSNR$ means that the error between the SR image and the original image is smaller, and the image quality is better; generally, a PSNR value above 30dB is considered as a high-quality image.

- $SSIM$(Structural Similarity Index)

$SSIM$ is a measure of the visual similarity of two images, which takes into account the brightness, contrast and structural information of the image, and can better reflect the human eye's perception of image quality than $PSNR$.It is calculated as follows:

$$SSIM_{(x,y)} = \frac{2(\mu_x\mu_y + c_1)(2\sigma_{xy} + c_2)}{(\mu_x{}^2\mu_y{}^2 + c_1)(2\sigma_x{}^2\sigma_y{}^2 + c_2)} \tag{7}$$

In the formula: $\mu_x$ and $\mu_y$ represent the brightness of the original image and the SR image at the (desired) pixel respectively; $\sigma_x$ and $\sigma_y$ are their mean brightness, and $\sigma_{xy}$ is their covariance; c are the small constant used to stabilize the division. $SSIM$ values range from 0 to 1, with values closer to 1 indicating that two images are more similar.

3) Results Presentation and Analysis

As shown in Figure. 8 and Table II, we display the original HR image set, the LR images captured in the actual scene, and the HR images reconstructed by SRGAN in the LR test set.

TABLE II: Comparison of Super-Resolution Methods

| Indicator | LR | 4×BICUBIC | 8×BICUBIC | Ours |
|---|---|---|---|---|
| PSNR | 28.97 | 30.57 | 32.37 | 34.11 |
| SSIM | 0.70 | 0.73 | 0.80 | 0.82 |

The data differences are evident. The BICUBIC method can improve image resolution but introduces a blurring effect, particularly at higher reconstruction coefficients, leading to edge detail loss in contact net wrist bolts. In contrast, the SRGAN model retains original image details, aligns better with human visual habits, and enhances the viewing experience. The data in Table II indicate significant improvements in key evaluation indicators, demonstrating that the reconstruction effect is more similar to the original image and closer to the real situation.

### C. Semantic Segmentation of Hexagonal Bolt Based on Improved Deeplabv3+ Model

1) Datasets Selection and Model Training

During the Deeplabv3+ model training phase, both HR and LR datasets were used, with the LR datasets serving as a control group. Based on SRGAN training results, we reconstructed the 280 test images in the LR datasets for subsequent testing. The Deeplabv3+ model was trained on datasets at different resolutions, using the same hardware configuration as the SRGAN training. Default epoch is 200, BatchSize is 32, initial learning rate is 0.007, momentum is set to 0.9, weight decay is 0.0001, and a cosine decay learning rate scheduling strategy controls the learning rate.

2) Evaluation Indicators

- mPA (Mean Pixel Accuracy)

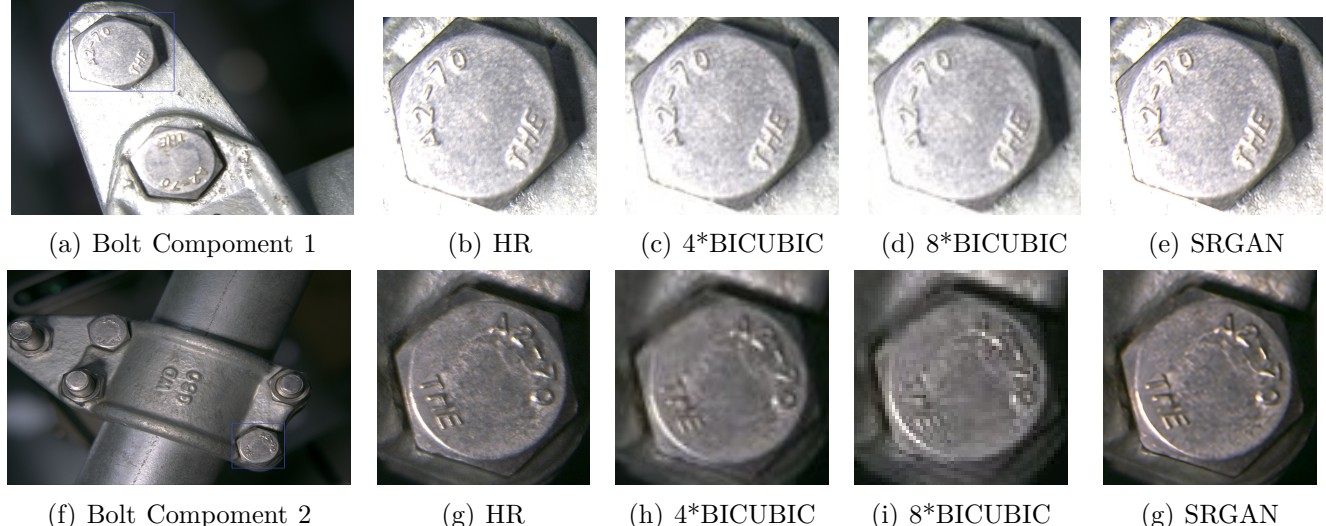

(a) Bolt Compoment 1    (b) HR    (c) 4*BICUBIC    (d) 8*BICUBIC    (e) SRGAN

(f) Bolt Compoment 2    (g) HR    (h) 4*BICUBIC    (i) 8*BICUBIC    (g) SRGAN

Fig. 8: Experimental visual results for Image Reconstruction.

Measures how accurately the model predicts each pixel class. Higher mPA values indicate better segmentation performance.

$$mPA = \frac{1}{K} \sum_{k=1}^{K} \frac{1}{I \times J} \sum_{i=1}^{I} \sum_{j=1}^{J} f(p_{ij} = k \wedge t_{ij} = k) \quad (8)$$

In the formula: $K$ is the number of segmentation categories; $I$ and $J$ denote the height and width of the image, respectively. $p_{ij}$ is the predicted pixel class at $(i,j)$; $t_{ij}$ is the actual pixel class at $(i,j)$; $f$ is the indicator function, which is 1 only when the function condition is met, that is, when both the predicted and actual classification of the pixel value are of class $k$, and 0 otherwise.

- mIoU (Mean Intersection over Union)

Measures the overlap between predicted and actual segmentation results. Higher mIoU values indicate better classification accuracy and segmentation network performance.

$$mIoU = \frac{1}{K} \sum_{i=0}^{K} \frac{p_{ii}}{\sum\limits_{j=0}^{K} p_{ij} + \sum\limits_{j=0}^{K} p_{ji} - p_{ii}} \quad (9)$$

In the formula: $K$ is the number of segmentation categories; $i$ $j$ represents the different classes; $p_{ij}$ is for predicting class $j$ as class $i$, which is 1 only if the classification is correct.

- Accuracy

Measures the fraction of correctly predicted pixels compared to the total number of pixels. Higher accuracy represents better semantic segmentation performance.

$$Accuracy = \frac{\sum\limits_{i=1}^{I} \sum\limits_{j=1}^{J} f(p_{ij} = t_{ij})}{I \times J} \quad (10)$$

In the formula: $I$ and $J$ represent the height and width of the image respectively; $p_{ij}$ is the predicted pixel class at $(i,j)$; $t_{ij}$ is the actual pixel class at $(i,j)$; $f$ is the indicator function, which is 1 only if the function condition is met, that is, if the predicted and actual classification of the pixel value are the same, and 0 otherwise.

3) Experimental Results and Analysis

As a crucial part of the method, the training of the semantic segmentation network rigorously demonstrates the correctness of the segmentation model and training method.

- Necessity of SRGAN reconstruction model:

To highlight the key role of incorporating image SR technology in our method, we chose the DeepLabv3+ model with MobileNetv2 as the backbone network and trained it on datasets with different resolutions. After training, different models were applied to the HR test image set reconstructed by SRGAN. The experimental results are as follows:

TABLE III: Performance Metrics for Different Data Sources

| Train and Test Data Source | $mPA$ | $mIoU$ | $Accuracy(\%)$ |
|---|---|---|---|
| Source 1 [a] | 97.92 | 96.26 | 99.44 |
| Source 2 [b] | 97.82 | 96.23 | 99.24 |
| Source 3 [c] | 95.91 | 94.50 | 98.19 |

[a] HR training set, HR test set reconstructed by SRGAN.
[b] LR training set, HR test set reconstructed by SRGAN.
[c] LR image training set, LR test set.

From the results in the Table III above, it can be concluded that compared with the training and testing on the original LR datasets, the three indicators of the DeepLabv3+ model have significantly improved after the introduction of image SR technology. Additionally, in the early dataset labeling stage, HR images are more

conducive to manual labeling compared to LR blurry images.

- Efficiency of the improved DeepLabv3+ model:

To highlight the innovation of the DeepLabv3+ model in our approach regarding the backbone network, several SS models were trained, including DeepLabv3+ using MobileNetv2 backbone network, the original DeepLabv3+, and the lightweight SS network PP-LiteSeg.

TABLE IV: Performance Metrics of Different Models

| Model | $mPA$ | $mIoU$ | $Accuracy(\%)$ |
|---|---|---|---|
| deeplabv3+-Mobilenetv2 | 97.92 | 96.26 | 99.44 |
| deeplabv3+-xception | 97.57 | 95.63 | 99.34 |
| PP-LiteSeg | 94.44 | 93.18 | 98.90 |

From the Table IV :further in-depth analysis of the different backbone networks of DeepLabv3+ reveals that when DeepLabv3+ is combined with MobileNetv2, the model reaches new heights in segmentation accuracy and overall performance. This combination not only optimizes the computational efficiency of the model but also significantly improves segmentation accuracy. In contrast, although DeepLabv3+ combined with Xception performs well, it is slightly worse on some performance metrics. The optimized model shows higher accuracy and efficiency in dealing with complex images and detail recognition, providing a solid technical foundation for the subsequent corner location task of hexagonal bolts.

### D. Determination of Corner Position of Hexagonal Bolt

1) Experimental procedure

The LR test images with a resolution of $512\times375$ are reconstructed by the SRGAN network, and then semantic segmentation is carried out by the DeepLabv3+-MobileNetv2 network. After multiple experiments, it is concluded that the best effect is achieved when the threshold of the LSD line segment detector is set to 0.25.

2) Evaluation index

Because the position information of the bolt corner is not clearly given, we manually marked the corner position using the labeling tool LabelMe as the position of the corner of our standard hexagonal bolt. The Euclidean distance formula is used to calculate the average distance error, and the formula is as follows:

$$Error = \frac{1}{6}\sqrt{(x^* - x)^2 + (y^* - y)^2} \qquad (11)$$

3) Experimental results and analysis

To demonstrate the accuracy of this method and the necessity of introducing image SR technology, the original method and this method are compared from two aspects: visual results and data results.

- Display of visual results:

Fig. 9 and Fig. 10 respectively show the results of hexagonal bolt edge detection using two different methods for

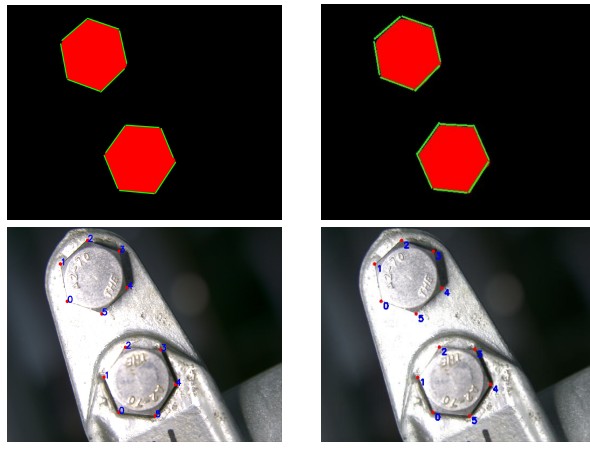

(a)Ours Method      (b)Low-Resolution

Fig. 9: Visual results for bolt compoment 1.

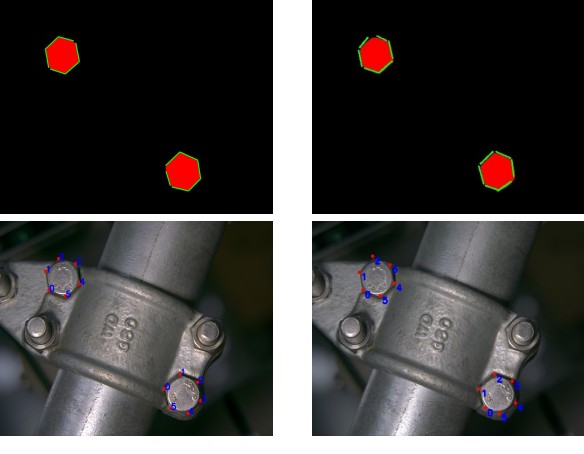

(a)Ours Method      (b)Low-Resolution

Fig. 10: Visual results for bolt compoment 2.

different catenary arm bolt components. Through careful comparison of these images, we observe that although the segmentation effect maintains integrity under LR, there are deficiencies in processing edge details. Specifically, the smoothness and uniformity of the edges have not reached the ideal state, and the corners of the hexagonal bolts are not visually sharp enough. These problems further affect the performance of the LSD line segment detection algorithm, preventing accurate encircling of hex bolt edges. Consequently, this imprecise edge detection affects the accuracy of corner detection, making the position accuracy fail to meet the expected requirements.

- Display of Data Results:

We tested the bolts (280×2=560 in total) under four different conditions (normal, low brightness, strong exposure, Gaussian noise) in the test set after SRGAN reconstruction and in the LR test set, respectively, and calculated the error value between the detected corner position of hexagonal bolts and the actual marked corner

TABLE V: The final data results.

| Method | Test Scenario | | | | Total Error |
|--------|--------|----------------|-----------------|----------------|-------------|
|        | Normal | Low Brightness | Strong Exposure | Gaussian Noise |             |
| Ours   | 2.38   | 2.86           | 1.46            | 2.12           | 2.21        |
| LR     | 3.95   | 4.25           | 2.79            | 3.45           | 3.61        |

position. The experimental data are as follows:

From Table V provided, we can clearly observe that in all experimental scenarios, the proposed method in this study shows significant improvement in error values compared to the original LR method. Specifically, the accuracy increases were 39.81%, 32.69%, 47.6%, and 38.6%, respectively. These data not only highlight the performance advantages of this method in various scenarios but also reflect its robustness under different conditions. When considering all cases comprehensively, the overall improvement of detection accuracy reaches 38.93%. This significant improvement greatly enhances the reliability and accuracy of corner position detection of hexagonal bolts, thus providing a more solid foundation for research and application in related fields.

## IV. Conclusion

In this paper, we introduce a method to determine the corner position of the arm bolt of a contact line by integrating image SR technology into the SS model. During maintenance, in the face of the challenge of LR images, we overcame the resolution limitation by generating corresponding HR images through the SRGAN model. On this basis, we combined the improved DeepLabv3+ model to complete the SS of hexagon bolts and accurately locate the corner points of the contact line arm bolts.

The reliability of the SRGAN reconstruction method is demonstrated by PSNR and SSIM values; the segmentation accuracy of the improved DeepLabv3+ model is verified by comparison experiments of mPA, mIoU, and Accuracy. Finally, the accuracy of the corner point position using the proposed method is proved to have improved by nearly 38.93% through distance calculation. The test results in various experimental scenarios show significant improvements, and the method is characterized by strong robustness.

Compared with the original LR method, the proposed method is more suitable for tightening the wrist bolt of the contact line under IBVS-based mechanical arm visual servoing by obtaining higher accuracy in the hexagon bolt corner position.

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
