# OpenReview forum: "Super-Resolution integrated Semantic Segmentation method for the Corner position of Catenary Bolt"
_IEEE.org/ICIST/2024/Conference — IEEE ICIST 2024 Conference Submission_

### Official Review · Reviewer_ep5N · 2024-08-24
**This article is very interesting, but there are still some shortcomings.**

**Rating:** 7
**Confidence:** 4

**Review:**

The conclusion chapter is also much too brief. I believe that the authors can have a relevant opinion regarding the usefulness of the model and what are its limitations.

---

### Official Review · Reviewer_ASgp · 2024-08-24
**Review Comments for Manuscript No. 17**

**Rating:** 7
**Confidence:** 3

**Review:**

1. There are several grammatical errors and some formatting issues in the text that need to be addressed. Additionally, the authors should ensure that the overall formatting of the manuscript complies with IEEE conference paper publication standards.

2. Can the proposed framework for an SR-integrated semantic segmentation method be applied to other fields or real-world applications? The authors are advised to discuss the potential applicability and future prospects of the method.

3. Is the choice of using SRGAN for super-resolution image reconstruction and the improved Deeplabv3+ for segmentation justified? Why were these specific models chosen over other possible alternatives? Additionally, how does the computational complexity of combining SRGAN with the improved Deeplabv3+ compare to other model combinations?

4. Why was high-precision Line Segment Detection (LSD) selected as the boundary detection method? What are the specific advantages of LSD in this application compared to other potential boundary detection methods, such as Canny edge detection or Hough Transform? The authors should clarify the rationale behind this choice.

---

### Decision · Program_Chairs · 2024-09-06

Accept (Oral)